# Marginal and Internal Precision of Zirconia Four-Unit Fixed Partial Denture Frameworks Produced Using Four Milling Systems

**DOI:** 10.3390/ma14102663

**Published:** 2021-05-19

**Authors:** Karl Martin Lehmann, Michael Weyhrauch, Monika Bjelopavlovic, Herbert Scheller, Henning Staedt, Peter Ottl, Peer W. Kaemmerer, Stefan Wentaschek

**Affiliations:** 1Department of Prosthodontics and Materials Science, University Medical Center of University of Mainz, Augustusplatz 2, 55131 Mainz, Germany; michiweyhrauch@googlemail.com (M.W.); monika.bjelopavlovic@unimedizin-mainz.de (M.B.); scheller@mail.uni-mainz.de (H.S.); stefan.wentaschek@unimedizin-mainz.de (S.W.); 2Department of Prosthodontics and Materials Science, University of Rostock, Strempelstr. 13, 18057 Rostock, Germany; henning@staedt.com (H.S.); peter.ottl@med.uni-rostock.de (P.O.); 3Department Life, Light & Matter, University of Rostock, Albert-Einstein-Straße 25, 18059 Rostock, Germany; 4Department of Oral- and Maxillofacial Surgery, University Medical Center of University of Mainz, Augustusplatz 2, 55131 Mainz, Germany; peer.kaemmerer@unimedizin-mainz.de

**Keywords:** marginal fit, internal fit, zirconia, FPD, CAD/CAM

## Abstract

Background: CAD/CAM systems enable the production of fixed partial dentures with small and reproducible internal and marginal gaps. Purpose: The purpose of this study was to evaluate the reproducibility of the marginal and internal adaptations of four-unit fixed partial denture frameworks produced using four CAD/CAM systems. Materials and Methods: Prepared dies of a master model that simulated the loss of the first left molar were measured. Fifteen frameworks were manufactured using four CAD/CAM systems (A–D). The internal fit was determined by the replica technique, and the marginal gap was determined by microscopy. ANOVA was carried out to detect significant differences, and the Bonferroni adjustment was performed. The global level of significance was set at 5%. Results: The mean gap size ranged from 84 to 132 µm (SD 43–71 µm). The CAD/CAM systems showed significant variance (*p* < 0.001), and system A (VHF) showed the smallest gaps. The smallest gaps for each system were in the molar part and in the marginal region of the frameworks (*p* < 0.001). Conclusions: The CAD/CAM systems showed significantly different gap sizes, particularly between premolars and molars and among the marginal, axial and occlusal regions. All of the systems are suitable for clinical application.

## 1. Introduction

All-ceramic materials are used widely in dentistry and have excellent biocompatibility and esthetics [1,2,3]. The demand for convenient, metal-free dentures has increased significantly in recent years. In particular, the excellent translucency of zirconia (ZrO2) in combination with an all-ceramic veneering material makes it suitable for fixed partial dentures (FPDs) [3,4,5]. Zirconia has a flexural strength from 900 to 1200 MPa and resists dispersion of cracks by undergoing a crystalline phase transformation [6,7,8]. Zirconia can be stabilized by yttrium oxide by sintering for up to 7.5 h at 1500 °C. With sintering, the dimensions of the restoration shrink from approximately 15% to 25% [8]. Due to its high stability, zirconia dental restorations can be cemented conventionally, unlike other all-ceramic materials.

With the development of computer-aided design (CAD)/manufacturing (CAM) technologies, which enable milling of pre-sintered zirconia, all-ceramic materials have become established in dentistry [8]. Compared to conventional techniques for producing all-ceramic dentures, CAD/CAM technologies have a simpler manufacturing process, because the design, for example, the construction of the connectors, can be made virtually predictable [7,8]. Furthermore, all-ceramic materials offer the potential for better aesthetics [1,4]. The fabrication of frameworks or copings range from single-tooth restorations to 14-unit bridges (Figure 1) [9]. Furthermore, in the meantime, CAD/CAM-based removable dentures have been produced [10].

The preparation is usually scanned. Therefore, different scanning systems are available, for example, Primescan (Sirona, Bensheim, Germany), CS3600 (Carestream, Rochester, Germany) or Trios 4 (3Shape, Kopenhagen, Denmark) for intraoral scanning and inEOS X5 (Sirona, Bensheim, Germany) or 3Shape (3Shape, Kopenhagen, Denmark) scanner for laboratory scanning procedures [11]. These scanning devices can also be used for removeable dental and craniofacial prothesis production [12,13] and based on the data, fixed partial dentures (FPDs) are designed using different software tools. After milling, the dentures must be sintered and fitted to the die [8,9,14,15].

The marginal and internal fit are important factors for the long-term wear of FPDs. Secondary caries and decementation or gingival irritation can result from an insufficient crown margin [16,17,18,19,20,21,22,23,24,25]. Furthermore, poorly fitted restorations render a tooth susceptible to pulpitis. If the internal fit is inadequate, strength and retention are reduced. The clinically acceptable marginal gap is reportedly 50 to 100 µm [15,16,17] or <120 µm [26,27]. Regarding this, there are basically different types of preparation, for example, knife edge finish line, bevel, chamfer, shoulder, shoulder bevel or modified shoulder/heavy chamfer preparation, whereas Beuer et al. reported the accuracy of fit as a function of preparation angle (4°, 8° and 12°) with the smallest gap for a 12° preparation angle [14]. However, as is well known, a larger angle goes hand in hand with less friction and a medium preparation angle is a good compromise. Therefore, it is important to use manufacturing systems that achieve small gaps of reproducible sizes.

In this study, we evaluated the marginal and internal fits of zirconia four-unit FPD frameworks produced using four milling systems, similar to the study by Vigolo and Fonzi [28].

The null hypothesis of this study is that there would be no difference in gap sizes among the CAD/CAM systems tested and no difference between molars and premolars or among the marginal, axial and occlusal regions using each system.

## 2. Materials and Methods

### 2.1. Master Model Preparation

A master model was manufactured (base metal alloy) to simulate loss of the first left mandibular molar, a frequent clinical situation. The premolars and the second molar were both prepared with a taper of 8° in a chamfer design to ensure an ideal combination of sufficient margin thickness and good visibility of the preparation margin to contain a bridge framework. The model was powdered (Dentaco Dentalindustrie und Marketing GmbH, Bad Homburg, Germany) and scanned using a Cerec system (Sirona Dental Systems GmbH, Bensheim, Germany).

### 2.2. CAD/CAM Design

FPDs were designed using Cerec three-dimensional (3D) software (Sirona, Bensheim, Germany) and Exocad (DentalCAD, Exocad GmbH, Darmstadt, Germany). A four-unit bridge framework was designed to replace the first lower molar (Figure 1). The default settings were a 10 µm cement spacer starting from 1 mm, a milling correction of 1.4 mm and an additional distance of occlusal from 0 x/y to 0.01 mm. Next, 15 duplicates of each pre-sintered and yttria-stabilized zirconia framework (Metoxit AG, Thayngen, Switzerland) were manufactured from the extended STL data set with the following milling systems: VHF CAM 4-02 Impression (dry milling) (exocad, System A, VHF Camfacture AG, Ammerbuch, Germany), Datron D5 (dry milling) (exocad, System B, Datron AG, Traisa, Germany), Cerec MCXL (wet milling) (Cerec inEos, System C, Sirona, Bensheim, Germany) and Roland DWX-50 (dry milling) (exocad, System D, Roland DG Deutschland GmbH Center, Willich, Germany) (Figure 2).

After milling, the frameworks were sintered at a maximum of 1450 °C for 9 h (inFire HTC, Sirona, Bensheim, Germany), and their fit was checked on the model.

### 2.3. Replica Technique

The internal gap was determined using the replica technique with a fluent polysiloxane impression material (Xantopren Comfort Light, Heraeus, Hanau, Germany). The silicone was mixed with this material, received into the framework crowns and placed on the master model with a pressure of 50 N using a dynamometer. After 5 min, the framework was carefully removed from the master model. A harder silicone material (Panasil Tray Soft, Kettenbach, Eschenburg, Germany) was mixed evenly and filled into the framework with the fluent silicone. The silicones were both removed from the framework crowns and cut in three planes in the oral-vestibular and mesial-distal directions. The internal gap was measured at 11 locations around the internal surface of each crown (Figure 3).

To minimize measurement inaccuracy, each section measurement was repeated five times using a 5× stereomicroscope (OPMI pico, Zeiss, Jena, Germany). To avoid device-specific measurement inaccuracies, the microscope unit was calibrated with software using a slide with a calibration circuit of 600 µm in diameter every 15 min. The marginal fit was recorded using the technique described by Holmes et al. [29].

### 2.4. Statistical Analysis

Data were imported into Statistical Package for the Social Sciences (SPSS) software (SPSS, Munich, Germany), and mean values were calculated and analyzed as descriptive statistics. One-way analysis of variance was carried out to detect significant differences among the systems in internal and marginal fit at the measurement locations. Bonferroni adjustment was used to control the family-wise error rate. The global level of significance was set at 5%.

## 3. Results

Table 1, Table 2, Table 3 and Table 4 show the marginal and internal gaps of the various regions and teeth produced using the four CAD/CAM systems.

### 3.1. Performance of the Systems

System A showed the smallest mean and maximum gap sizes and the smallest standard deviation, followed by systems D, B and C (*p* < 0.001, Table 1).

### 3.2. Gap Size According to Tooth Type

The gap size of Tooth 37 was significantly smaller than that of Tooth 34 for systems B, C and D, and likewise the gap size of Tooth 37 was smaller than that of Tooth 35 for systems B and D. The gap sizes of Teeth 34 and 35 differed significantly between systems B and D (all *p* < 0.001, Table 2 and Table 3).

### 3.3. Gap Size According to Region

Systems A, B and C showed a significantly smaller gap size in the marginal area of the framework as compared with the axial and occlusal regions (*p* < 0.001). The occlusal area showed the largest internal gaps with all four systems. The mean gap sizes in the axial wall were between those in the other areas (*p* < 0.001). System A showed the smallest gap sizes in all three regions, with the exception of system D (*p* < 0.001, Table 4).

## 4. Discussion

Marginal adaption is an important means of enhancing tenacity [30,31,32,33]. An enlargement of the marginal opening determines the clinical risk of secondary caries and inflammation of the adjacent tissue or the tooth itself. Therefore, it is important that systems produce small and reproducible marginal and internal gaps [31,32,33]. It can be difficult to standardize parameters, especially those of CAD/CAM systems. These systems typically have varying standards for scanning and grinding. Moreover, the software packages may enable only a few parameters to be changed. Thus, these systems cannot be compared directly in terms of marginal and internal fit. However, a small and reproducible gap size (in terms of internal and marginal fit) is important to avoid periodontal disease and secondary caries. Therefore, the internal and marginal fit was assessed [18,20,34], with the hypothesis that there would be no difference in gap sizes milled using the four systems or among the axial, occlusal, and marginal regions of FPDs.

We found significant differences among the systems (84–101 µm), as well as among teeth (Tooth 34, 83–141 µm; Tooth 35, 85–143 µm; Tooth 37, 85–113 µm) and regions (marginal, 7–49 µm; axial, 24–117 µm; occlusal, 20–165 µm), using each system. There are numerous potential reasons for these differences; there are many differences in the construction of the milling apparatus of these CAD/CAM systems as well as the software applications, with different setting options. Therefore, varying numbers of axes and grinding strategies may explain the differences [33,34,35]. Systems A, B and D are dry-milling units, whereas system C is a wet-milling unit. System A showed the smallest gaps with the smallest standard deviation. Within system A, there was no significant difference among the teeth, indicating consistently small gaps independent of the framework geometry. In contrast, systems B, C and D showed a smaller gap with tooth 37 as compared with teeth 34 and 35, possibly because of the larger radius of tooth 37. This led to higher precision because the grinding wheels of the smallest burs had more space in which to work and the larger die may have been easier to scan. Indeed, differences have been reported in the marginal and internal fit among different dies [34] and molars typically have smaller gaps than premolars [32].

The occlusal area had greater space than the axial wall and the marginal region, as reported by Reich et al., who also confirmed that the marginal gap was smaller than the other areas of the die [22]. Beuer et al. also showed that the occlusal area of zirconia three-unit FDPs had a larger gap than the chamfer area [19]. This may have been due to the grinding strategy, milling process [32,36,37,38], setting options and/or the manual adjustment process. Greater marginal adaption is intended to achieve marginal sealing after fixation of the FPD using cement or adhesive materials, avoiding secondary caries and periodontal disease. In this respect, a small marginal gap is important for longevity, but an uneven contact situation with punctual contacts instead of a uniform support, especially at the marginal edge, can lead to tensions in the framework and veneering ceramic layer. With respect to the material-related properties of zirconia materials as compared with metal-based restorations, it is also of considerable relevance to achieve a high marginal adaption to take the complete interface with the fixation materials into account. In addition, it is unknown to what extent the size of the marginal gap and the type and quality of fixation material influence restoration longevity. Moreover, it is important to change frequency of cutting burs to achieve small gaps and predictable outcomes for dental CAD/CAM prostheses [39]. Furthermore, whether smaller gaps or the fixation protocol are most important for longevity remains to be determined. Thus, the four CAD/CAM systems were not comparable in terms of the marginal and internal gaps at different locations and according to the tooth. Additionally, different combinations of software and hardware components yielded different results for internal and marginal gaps, as in prior studies [22,27,37,38]. It remains to be considered whether, in the future, 3D printing methods will also generate clinically acceptable prothesis and be relevant in addition to milling [40].

Therefore, the working hypothesis, i.e., that there would be no difference in the internal and marginal gaps of restorations produced using the CAD/CAM systems, was rejected. However, it was not possible to reach conclusions on the clinical impact of the target parameters.

## 5. Conclusions

For FPDs, the four CAD/CAM systems showed significantly different gap sizes between premolars and molars and among the marginal, axial and occlusal regions. Thus, the abutment tooth geometry seems to influence the gap situation. The systems are suitable for clinical applications. The data collected in this study require verification in clinical practice.

## Figures and Tables

**Figure 1 materials-14-02663-f001:**
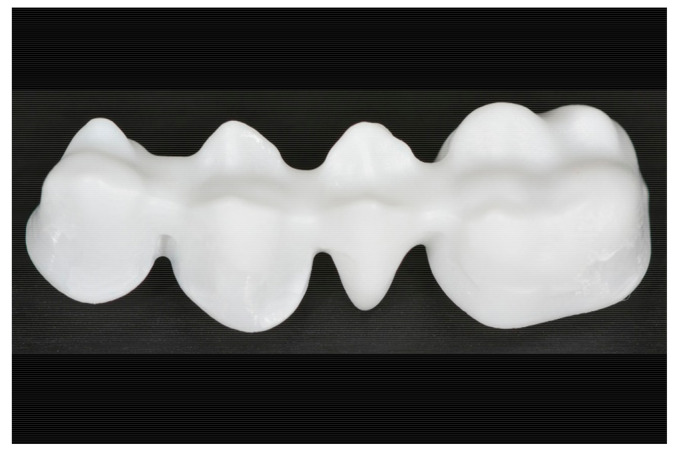
Zirconia four-unit framework.

**Figure 2 materials-14-02663-f002:**
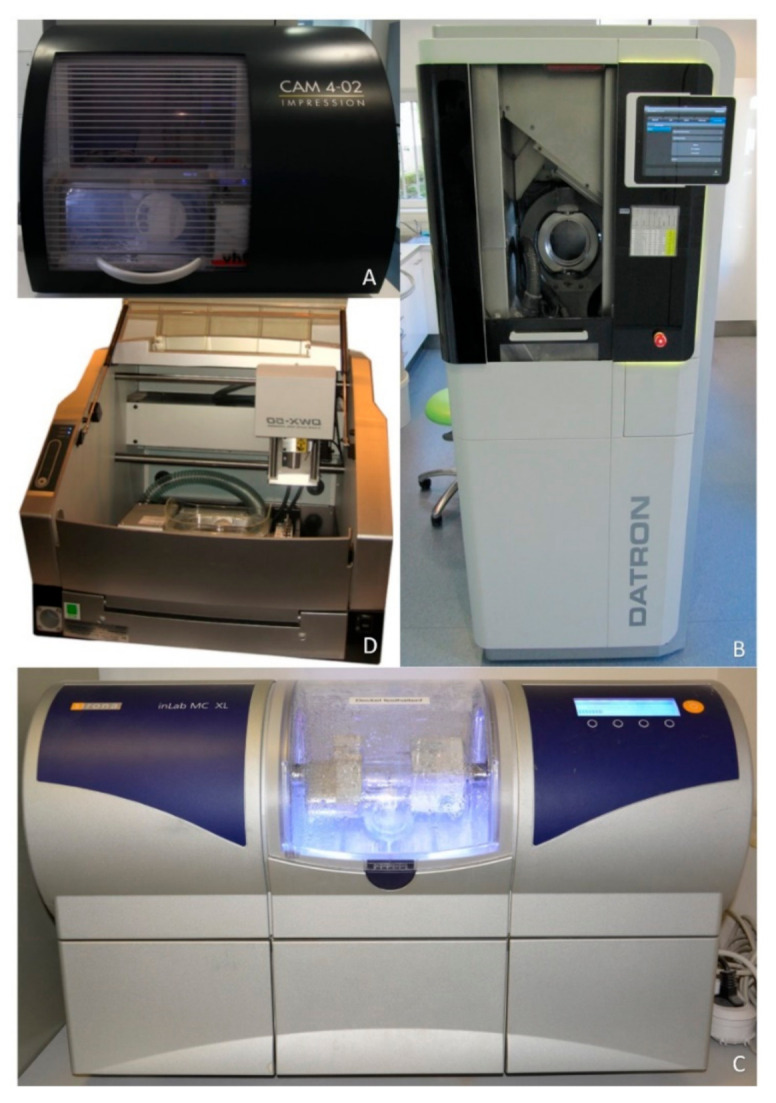
CAM Systems. System (**A**) (VHF CAM 4-02 Impression), system (**B**) (Datron D5), system (**C**) (Cerec MCXL) and system (**D**) (Roland DWX-50).

**Figure 3 materials-14-02663-f003:**
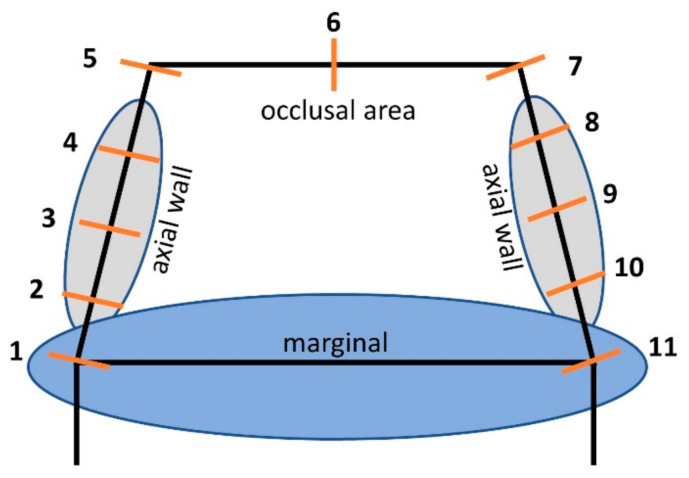
Measurement points at the marginal and internal surfaces of a restoration.

**Table 1 materials-14-02663-t001:** Performance of the CAD/CAM systems.

System	Mean (µm)	SD	Min (µm)	Max (µm)
**A**	84	43	0	285
**B**	113	65	0	433
**C**	132	71	5	579
**D**	101	67	10	427

**Table 2 materials-14-02663-t002:** Gap sizes of FPDs according to the tooth and the CAD/CAM system (µm).

Tooth	(A/B/C/D)	(A/B/C/D)	(A/B/C/D)	(A/B/C/D)
	Mean	SD	Min	Max
34	83/109/141/110	45/66/80/74	0/0/5/10	285/428/579/416
35	85/125/143/105	44/72/69/68	0/8/11/10	256/433/551/427
37	85/104/113/88	41/55/58/55	0/0/8/11	223/300/505/339

**Table 3 materials-14-02663-t003:** Repeated-measures one-way analysis of variance results.

Tooth	(A/B/C/D)	(A/B/C/D)	(A/B/C/D)
	34	35	37
34	-/-/-/-	0.107/<0.001/1.000/<0.001	0.125/<0.001/<0.001/<0.001
35		-/-/-/-	1.000/<0.001/1.000/<0.001
37			-/-/-/-

**Table 4 materials-14-02663-t004:** Mean gap size with standard deviation, minimum and maximum, according to region (µm).

Region	A	B	C	D
Marginal (Q)	38/7/24/49	72/25/42/125	64/19/39/99	58/27/32/104
Axial wall (R)	76/24/44/117	94/40/53/167	141/29/95/226	75/25/39/135
Occlusal area (S)	132/20/89/165	178/29/138/250	159/27/105/195	182/37/134/269

## Data Availability

Data sharing is not applicable.

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
