# Peer review of "Marginal and Internal Precision of Zirconia Four-Unit Fixed Partial Denture Frameworks Produced Using Four Milling Systems"

_materials, 2021, doi:10.3390/ma14102663_

Round 1

Reviewer 1 Report

In the abstract section do not generally indicate system A but name it specifically otherwise it is not clear to the reader

-Remove replica technique from keywords

-line 47-48 sentence too general. Indicate, by taking bibliographic references, the advantages and disadvantages of the CAD CAm technology compared to the traditional one, in terms of design, construction, connectors, etc.

-line 57 when it comes to scanning the preparation, some references to the various intraoral scanning systems must be made, especially in terms of accuracy and precision. In this regard, I recommend that you insert the following scientific work in the reference section, which could be of help to readers:

Pagano S, Moretti M, Marsili R, Ricci A, Barraco G, Cianetti S. Evaluation of the Accuracy of Four Digital Methods by Linear and Volumetric Analysis of Dental Impressions. Materials (Basel). 2019 Jun 18; 12 (12): 1958.

-LINE 66 EXPRESSLY INDICATE "THE NULL HYPOTHESIS OF THE STUDY IS ... .."

-Line 77 which finishing margin has been selected for the abutments? What was the rationale for the choice? There are numerous studies in the literature which, in fact, correlate the prosthesis abutment marginal gap with the prosthetic finishing margin. Describe these aspects in the introduction section

-How were the 4 procedural systems selected? Indicate, perhaps in a table, the main characteristics of the various systems, and indicate which were the selection criteria used

-There is, in my opinion, a problem related to the abbreviations: A-D are not helpful to the reader, just as the names are included in table 4; the nomenclature must be homogeneous and I recommend replacing it perhaps with an acronym that helps identification; this part of the study appears confused

-The discussion section appears too small and focused exclusively on the results of the study. An aspect to be argued would be, for example, the possible role of CAD / CAM technology in prostheses in general, not just the fixed one. Recent developments, in fact, are also taking place in the removable prosthesis.

-The reference section should be updated with more recent works. A topic as innovative as CAD / CAM deserves a decidedly more recent bibliographic analysis

Author Response

In the abstract section do not generally indicate system A but name it specifically otherwise it is not clear to the reader - we add the Information VHF...

-Remove replica technique from keywords - we agree and remove replica technique from keywords

-line 47-48 sentence too general. Indicate, by taking bibliographic references, the advantages and disadvantages of the CAD CAm technology compared to the traditional one, in terms of design, construction, connectors, etc. - So we add "Compared to conventional techniques for producing all-ceramic dentures, CAD/CAM technologies have a simpler manufacturing process, because the design, e.g. the construction of the connectors can be made virtual predictable. Further all-ceramic materials offer the potential for better aesthetics."

-line 57 when it comes to scanning the preparation, some references to the various intraoral scanning systems must be made, especially in terms of accuracy and precision. In this regard, I recommend that you insert the following scientific work in the reference section, which could be of help to readers:

Pagano S, Moretti M, Marsili R, Ricci A, Barraco G, Cianetti S. Evaluation of the Accuracy of Four Digital Methods by Linear and Volumetric Analysis of Dental Impressions. Materials (Basel). 2019 Jun 18; 12 (12): 1958.

We insert the reference recommended and add "Therefore different scanning systems are available, e.g. Primescan, CS 3600 or Trios 4 for intaoral scanning and inEOS X5 or 3Shape-Scanner for laboratory scanning procedures. Those scanning devices can also be used for removeable dental and craniofacial prothesis production."  

-LINE 66 EXPRESSLY INDICATE "THE NULL HYPOTHESIS OF THE STUDY IS ... .." - We corrected "The null hypothesis of this study is that there would be no difference in gap sizes among the CAD/CAM systems tested and no difference between molars and premolars or among the marginal, axial, and occlusal regions using each system."

-Line 77 which finishing margin has been selected for the abutments? What was the rationale for the choice? There are numerous studies in the literature which, in fact, correlate the prosthesis abutment marginal gap with the prosthetic finishing margin. Describe these aspects in the introduction section - we add: "Regarding this there are basically different types of preparation, e.g. knife edge finish line, bevel, chamfer, shoulder, shoulder-bevel or modified shoulder/heavy chamfer preparation, whereas Beuer et al reported the accuracy of fit as a function of preparation angle (4°, 8° and 12°) with the smallest gap for 12° preparation angle. However, as is well known, a larger angle goes hand in hand with less friction, so a medium preparation angle is a good compromise."...."Both premolars and the second molar were prepared with a taper of 8° in a chamfer design to ensure an ideal combination of sufficient margin thickness and good visibility of the preparation margin to contain a bridge-framework"

-How were the 4 procedural systems selected? Indicate, perhaps in a table, the main characteristics of the various systems, and indicate which were the selection criteria used - We have focud on that in the production process the scanning and software part stayed the same and only the manufacturing process differed. So we chose these systems because they had a high level of market penetration.

-There is, in my opinion, a problem related to the abbreviations: A-D are not helpful to the reader, just as the names are included in table 4; the nomenclature must be homogeneous and I recommend replacing it perhaps with an acronym that helps identification; this part of the study appears confused - So we agree and corrected the table 4 (also with SD, Minimum, Maximum).

-The discussion section appears too small and focused exclusively on the results of the study. An aspect to be argued would be, for example, the possible role of CAD / CAM technology in prostheses in general, not just the fixed one. Recent developments, in fact, are also taking place in the removable prosthesis. - So we agree and add more information in the introduction and the discussion part: "Further in the meantime, CAD/CAM-based removable dentures can are be produced.", "Moreover it is important to changing frequency of cutting burs to achieve small gaps and predictable outcomes for dental CAD/CAM prostheses." and "It remains to be considered that in the future, 3D printing methods will also generate clinical acceptable prothesis and be relevant in addition to milling."

The reference section should be updated with more recent works. A topic as innovative as CAD / CAM deserves a decidedly more recent bibliographic analysis - We agree and insert more recent studies:

10. Tanveer, W.; Ridwan-Pramana, A.; Molinero-Mourelle, P.; Koolstra, J.H.; Forouzanfar T. Systematic Review of Clinical Applications of CAD/CAM Technology for Craniofacial Implants Placement and Manufacturing of Nasal Prostheses 2021 Int J Environ Res Public Health Apr 3;18(7):3756

11.    Pagano, S.; Moretti, M.; Marsili, R.; Ricci, A.; Barraco, G.; Cianetti, S. Evaluation of the Accuracy of Four Digital Methods by Linear and Volumetric Analysis of Dental Impressions. 2019 Materials (Basel) Jun 18; 12 (12): 1958.

12.    Lee, S.J.; Jamjoom, F.Z.; Le, T.; Radics, A.; Gallucci, G.O. A clinical study comparing digital scanning and conventional impression making for implant-supported prostheses: A crossover clinical trial 2021 J Prosthet Dent. Feb 15:S0022-3913(21)00028-7.

13.    Knechtle, N.; Wiedemeier, D.; Mehl, A.; Ender, A. Accuracy of digital complete-arch, multi-implant scans made in the edentulous jaw with gingival movement simulation: An in vitro study. 2021 J Prosthet Dent. Feb 18:S0022-3913(21)00019-6.

         14.    Batak, B.; Cakmak, G.; Seidt, J.; Yilmaz, B. Load to failure of high-density polymers for implant-supported fixed, cantilevered prostheses with titanium bases 2021 Int J Prosthodont Feb 19. doi: 10.11607/ijp.7036.

40. Song, D.B.; Han, M.S.; Kim, S.C.; Ahn, J.; Im, Y.W.; Lee, H.H. 2021 Influence of Sequential CAD/CAM Milling on the Fitting Accuracy of Titanium Three-Unit Fixed Dental Prostheses. Materials (Basel). Mar 13;14(6):1401.

          41.    Herpel, C; Tasaka, A.; Higuchi, S.; Finke, D.; Kühle, R.; Odaka, K.; Rues, S.; Lux, C.J.; Yamashita, S.; Rammelsberg, P.; Schwindling, F.S. 2021 Accuracy of 3D Printing Compared with Milling - a Multi-Center Analysis of          Try-In Dentures. J Dent. 2021 Apr 24:103681.

Reviewer 2 Report

1. The summary must be drawn up in the form required by the template, without alignment, etc.

2. Bibliography, figures, and so on must comply with the format required by the template.

3. In the material part there is not enough comprehensive data about the structure and provenance of the materials used in the study.

4. The results part must be redone with the introduction of both statistical processing data and interpretation of the results obtained.

5. The part of the conclusions needs to be rewritten in order to provide more relevant data.

Author Response

The summary must be drawn up in the form required by the template, without alignment, etc. We checked this.

2. Bibliography, figures, and so on must comply with the format required by the template. We checked this.

3. In the material part there is not enough comprehensive data about the structure and provenance of the materials used in the study.- We add for the zirconia "Next, 15 duplicates of each presintered and yttrium-stabilized zirconia framework (Metoxit AG, Thayngen, Switzerland) were manufactured from the extended STL data set with the following milling systems:....". Furter we ad the kind of milling modus (wet/dry): "VHF CAM 4-02 Impression (dry milling) (VHF Camfacture AG, Ammerbuch, Germany; exocad; System A), Datron D5 (dry milling) (Datron AG, Traisa, Germany; exocad; System B), Cerec MCXL (wet milling) (Sirona Dental Systems; Cerec inEos; System C), and Roland DWX-50 (dry milling) (Roland DG Deutschland GmbH Center, Willich, Germany; exocad; System D) (Fig. 2).

      4. The results part must be redone with the introduction of both statistical processing data and interpretation of the results obtained. We checked the statistical evaluation again. We do not really know what"redone"                means. If possible please specify the request.

5. The part of the conclusions needs to be rewritten in order to provide more relevant data. We add the sentence "Thus the abutment tooth geometry seems to influence the gap situation". If possible please specify the request.

Round 2

Reviewer 1 Report

all comments were added

Author Response

Dear Reviewer,

thank you for your time and your helpfulness. The document was sent to an proofreading for english language. We hope it meets your expections.

Best regards

Reviewer 2 Report

  1. Abstract is not in accord with the template https://www.mdpi.com/files/word-templates/materials-template.dot "A single paragraph of about 200 words maximum."
  2. References is not in accord with the template "References should be numbered .. [1] or [2,3], or [4–6], ..".
  3. "All figures and tables should be cited in the main text as Figure 1, Table 1, etc..."
  4. In Materials and Methods statistical analysis is presented, but in Results it is not used this data from pointe of calculated and analyzed...This must be put in accord with line 230 to 243.

Author Response

Dear Reviewer,

thank you for your time and helpfulness. We reduced the words in the abstract (now there are 200 words). Further we changed the style for the references in the text to [X]. Beyond that we do not use the abbreviations for Figures and tables in the text and we added informations of the results in the discussion section. So the reader have the connection between the results of this study and discussed literature. Thus we hope to meet your expections. 

Best regards

This manuscript is a resubmission of an earlier submission. The following is a list of the peer review reports and author responses from that submission.